## Research Article

global mental health; traditional healer; faith based healers; faith healing; health care system

**Corresponding author:**
Yang Jae Lee;
Email: yangjae.lee@yale.edu

# Perspectives of traditional healers, faith healers, and biomedical providers about mental illness treatment: qualitative study from rural Uganda

Yang Jae Lee[1,2] , Mary Coleman[3], Kayera Sumaya Nakaziba[2], Nicole Terfloth[3], Camryn Coley[3], Anurag Epparla[3], Nolan Corbitt[3], Rauben Kazungu[2], Job Basiimwa[2], Corinne Lafferty[4], Kassidy Cole[5], Grace Agwang[6], Emrose Kathawala[3], Travor Nkolo[7], William Wogali[7], Egessa Bwire Richard[7], Robert Rosenheck[1] and Alexander C. Tsai[8,9,10]

[1]Department of Psychiatry, Yale University, New Haven, CT, USA; [2]Empower Through Health, Iganga, Uganda; [3]College of Medicine, University of Tennessee Health Science Center, Memphis, Tennessee, USA; [4]The Ohio State University, Columbus, OH, USA; [5]Notre Dame University, South Bend, IN, USA; [6]Uganda Christian University, Mukono, Uganda; [7]Cavendish University, Kampala, Uganda; [8]Center for Global Health and Mongan Institute, Massachusetts General Hospital, Boston, MA, USA; [9]Department of Psychiatry, Harvard Medical School, Cambridge, MA, USA and [10]Department of Psychiatry, Mbarara University of Science and Technology, Mbarara, Uganda

## Abstract

Most people with mental illness in low and middle-income countries (LMICs) do not receive biomedical treatment, though many seek care from traditional healers and faith healers. We conducted a qualitative study in Buyende District, Uganda, using framework analysis. Data collection included interviews with 24 traditional healers, 20 faith healers, and 23 biomedical providers, plus 4 focus group discussions. Interviews explored treatment approaches, provider relationships, and collaboration potential until theoretical saturation was reached. Three main themes emerged: (1) Biomedical providers' perspectives on traditional and faith healers; (2) Traditional and faith healers' views on biomedical providers; and (3) Collaboration opportunities and barriers. Biomedical providers viewed faith healers positively but traditional healers as potentially harmful. Traditional and faith healers valued biomedical approaches while feeling variably accepted. Interest in collaboration existed across groups but was complicated by power dynamics, economic concerns, and differing mental illness conceptualizations. Traditional healers and faith healers routinely referred patients to biomedical providers, though reciprocal referrals were rare. The study reveals distinct dynamics among providers in rural Uganda, with historical colonial influences continuing to shape relationships and highlighting the need for integrated, contextually appropriate mental healthcare systems.

## Impact statement

This study provides critical insights into the complex relationships between traditional healers, faith healers, and biomedical providers in rural Uganda, offering a nuanced understanding of mental health care delivery. By revealing the willingness of traditional and faith healers to collaborate with biomedical providers – rather than treating them as a homogenous group – the research offers a nuanced understanding of how different healing approaches interact within the local healthcare ecosystem.

The findings highlight the intricate dynamics of mental healthcare in rural Uganda, demonstrating that collaboration is neither straightforward nor uniform. While traditional healers and faith healers both showed a willingness to refer patients to biomedical providers, their motivations, approaches, and relationships with the formal healthcare system varied greatly. Faith healers occupied a more complementary position, while traditional healers maintained a more contested stance within the healthcare landscape.

The research also sheds light on the lasting impact of colonial-era attitudes on current healthcare dynamics, offering a deeper understanding of the socio-cultural context that shapes mental healthcare delivery in Uganda and potentially other post-colonial African nations.

These findings suggest that developing effective mental healthcare integration requires nuanced, tailored approaches. For faith healers, who already occupy a more complementary role and share similar spiritual frameworks with biomedical providers, more integrated referral and communications systems may be appropriate. In contrast, collaboration with traditional healers would require more fundamental approaches that respect their cultural expertise and address potential economic marginalization.





Ultimately, the study offers a pathway to more nuanced approaches to mental healthcare in rural Uganda and similar contexts across Africa. By identifying both barriers and opportunities for collaboration, this study provides valuable guidance for developing integrated care models that could improve mental health access and outcomes in rural Uganda and similar settings across Africa.

## Introduction

The delivery of mental healthcare occurs within complex historical and cultural contexts where multiple healing traditions coexist. In Uganda, as in many post-colonial nations, mental healthcare involves an interplay between biomedical approaches, traditional healing practices, and faith-based care (Sundararajan et al., 2020; Teuton et al., 2007). While epidemiological data suggest that only 29% of people with psychotic disorders in low and middle-income countries (LMICs) receive treatment (World Health Organization, 2021), this framing of a "treatment gap" requires careful examination, as it privileges Western psychiatric paradigms over local healing systems that many find meaningful and effective (Abbo, 2011; Ensink and Robertson, 1999; Gbadamosi et al., 2022; Gureje et al., 2020; Gureje et al., 2015; Lambert et al., 2020; Sorsdahl et al., 2009; Teuton et al., 2007).

These parallel care systems reflect different conceptualizations of mental illness and healing. In many Ugandan communities, mental distress may be understood through spiritual, social, or ancestral frameworks that differ from biomedical disease models. Traditional healers, including herbalists and spirit mediums, often approach mental illness through these cultural frameworks, while faith healers – predominantly Christian and Muslim in post-colonial Uganda – blend spiritual and psychological approaches (Akol et al., 2018b). Biomedical providers, trained in Western psychiatric traditions, emphasize neurobiological and psychological frameworks (Abbo et al., 2019).

Understanding how these different healing traditions interact is crucial for developing culturally appropriate mental healthcare systems. Previous research suggests approximately 50% of patients in African countries seek care from traditional healers and faith healers before turning to biomedical providers (Burns and Tomita, 2015a), often continuing to engage multiple systems simultaneously (van der Watt et al., 2018). This pluralistic care-seeking reflects both practical considerations –traditional healers and faith healers are often more accessible in rural areas, providing culturally appropriate care and psychosocial support (Campbell-Hall et al., 2010; Nortje et al., 2016; Sorsdahl et al., 2009; van der Watt et al., 2018; van der Zeijst et al., 2023) – and the perceived complementarity of different approaches (van der Watt et al., 2018).

This complex healthcare landscape has prompted increasing interest in collaborative care models. The World Health Organization (WHO) recognizes the potential role of traditional healers and faith healers within broader mental healthcare systems, though implementation remains challenging (Akol et al., 2018b; World Health Organization, 2013). Early evidence from Nigeria and Ghana suggests that structured collaboration between different types of providers can improve outcomes while remaining cost-effective (Gureje et al., 2020; Ofori-Atta et al., 2018). However, implementing such collaboration faces numerous challenges. Previous research indicates that while traditional healers and faith healers often express a willingness to work with biomedical providers, this openness is not always reciprocated (Campbell-Hall et al., 2010; Meissner, 2004). Barriers include differences in beliefs about mental illness causation and

treatment, concerns about exploitation, and economic considerations (Akol et al., 2018b; Meissner, 2004).

This study examines the attitudes and perspectives of traditional healers, faith healers, and biomedical providers in the rural Buyende District in Uganda regarding mental health treatment and potential collaboration. We aim to understand the barriers to and opportunities for developing effective collaborative models of mental health care in this context. Our findings will contribute to the growing literature on mental healthcare delivery in LMICs and provide insights that could inform policy and practice in similar settings across sub-Saharan Africa (Esan et al., 2019; Gureje et al., 2020; Pham et al., 2021; Solera-Deuchar et al., 2020; van der Watt et al., 2017).

## Methods

### Study site

This qualitative exploratory study was conducted in the Buyende District, a rural area of eastern Uganda with a population of approximately 450,000 people (Uganda Bureau of Statistics, 2017). Buyende District was selected for this study because it is representative of many rural areas in Uganda. The majority of the working population is engaged in subsistence farming (Uganda Bureau of Statistics, 2022). Previous research in this region has highlighted the complex healthcare-seeking behaviors of local residents, including the use of both traditional and biomedical care options (Lee et al., 2019a; Lee et al., 2019b). Specifically, providers were interviewed from the Irundu (approximately 26,000 people), Kagulu (approximately 58,000 people), and Bukutula (46,500) sub-counties (Uganda Bureau of Statistics, 2021).

### Study design and data collection

We employed both in-depth interviews (IDIs) and focus group discussions (FGDs) to capture complementary perspectives: IDIs allowed for detailed exploration of individual experiences and sensitive topics, while FGDs enabled observation of group dynamics and consensus-building around collaboration possibilities. Data collection occurred between May 2023 and August 2023.

We recruited a purposive sample of traditional healers, faith healers, and biomedical providers. The study was explained to the local leadership and community health workers, who then assisted in identifying traditional healers and faith healers. Biomedical providers were sampled from a roster of health centers in the Buyende District. Interviews were conducted in private settings of the participants' choosing, typically their homes or workplaces. Traditional healers employed various combinations of herbal remedies, spiritual practices, and cultural healing traditions, while faith healers were predominantly from Christian and Islam denominations. Biomedical providers were primary healthcare workers working in local health centers and consisted of nurses, midwives, and clinical officers. The extent of experience with mental health problems may have been variable across the providers. No

participant identified as belonging to more than one of the three categories of participants. Participants were required to be 18 years older. Eligibility for traditional healers and faith healers required seeing an average of five or more patients for mental illness per month in the past three months. Biomedical providers needed at least 1 year of post-secondary medical training.

Altogether, 53 in-depth interviews (IDIs) were conducted, comprising 18 traditional healers, 18 faith healers, and 17 biomedical providers. Afterward, four focus group discussions (FGDs) were organized, with six to seven participants in each group. Two FGDs consisted of traditional healers, totaling 13 participants in total (seven of whom also participated in the IDIs, and six who did not) to capture the range of healing practices within this group. Another FGD involved seven faith healers (five of whom also participated in IDIs, and two who did not), while the fourth FGD comprised six health workers, none of whom participated in IDIs. In total, the study gathered distinct perspectives from 24 traditional healers, 20 faith healers, and 23 biomedical providers (n=67). Data collection continued until theoretical saturation was reached, which we defined as three consecutive interviews yielding no new themes or substantial insights. This was assessed through regular team discussions of emerging findings.

Research assistants with backgrounds in social science and/or public health conducted the interviews after completing training in qualitative methods and research ethics. All research assistants interviewing participants were either Ugandan master's or bachelor's students. All interviewers were fluent in both English and Lusoga (local language). For some interviews, American research assistants (either M.D., MPH, or bachelor's level students) observed and assisted with audio recording but did not directly participate in the interviews. Interviews used semi-structured guides with probing questions developed through literature reviews and pilot testing. Key topics included attitudes toward other providers, treatment approaches, and views on collaboration. Each participant was approached in the field by research assistants who spoke the local language (Lusoga) and a community health worker who requested their participation in the study. Each participant was read a consent form describing the aim of the study and any potential risks and discomforts to request verbal and written consent. The consent forms were translated from English into Lusoga by native speakers and back-translated to English by a different Lusoga native speaker to verify fidelity to the original. If participants could not sign their names, a fingerprint was obtained, and an impartial witness was asked to sign the consent form.

IDIs lasted between 30 and 60 min. While FGDs lasted approximately 90 min on average. All interviews were audio-recorded, transcribed, and translated into written English. Data collection proceeded until content saturation was achieved.

## Data analysis

We used the framework method to inductively identify recurring themes in the data (Gale et al., 2013). This method involves systematically coding the data and organizing it into a matrix to facilitate comparison across cases. To ensure rigor, we triangulated data by comparing findings between IDIs and FGDs, using multiple coders with different cultural perspectives, and verifying interpretations through team discussions. Each interview transcript was reviewed by two to three study team members, who familiarized themselves with the content and applied a paraphrase, or code, to passages that they interpreted as important. Study team members

first coded the first 3 to 5 transcripts of each respondent group separately (in groups of 2 or 3). Each group consisted of at least one Ugandan and one American. When questions arose about cultural or linguistic interpretation during coding, the team consulted with the Ugandan researchers and original interviewers to ensure an accurate understanding of participants' meanings in their cultural context. The study team members then met to compare the codes and agreed on codes to apply to all subsequent transcripts. Study team groups then coded the remaining transcripts separately.

Codes were grouped into clearly defined categories, forming the analytical framework. The analytical framework was edited as additional codes emerged from subsequent transcripts. Finally, the analytical framework was converted into themes and sub-themes. This conversion process involved reviewing the categories in the analytical framework, identifying overarching patterns or concepts that connected multiple categories, and organizing these into broader themes. Sub-themes were then developed to capture more specific aspects or variations within each main theme. This hierarchical organization allowed us to present our findings in a coherent and meaningful structure, highlighting the major insights from our data while preserving the nuances and complexities revealed in our analysis. No qualitative data analysis software was used.

## Ethical considerations

This research study was approved by the Institutional Review Boards of The AIDS Support Organization, Uganda, and Yale University. In accordance with national guidelines, we received approval to conduct the study from the Uganda National Council of Science and Technology.

## Results

We inductively identified three themes from the data (Table 1) illustrating the complex relationships between different types of healthcare providers in rural Uganda. These themes captured how historical power dynamics, economic realities, and different conceptualizations of mental illness intersect to shape current and potential collaboration between providers. A visual representation of the relationship among the three providers can be found in Table 2.

***Theme 1:*** *Biomedical providers' perspective on traditional healers and faith healers*

Biomedical providers drew sharp distinctions between traditional healers and faith healers, revealing deeply rooted attitudes about legitimacy and trust in mental healthcare delivery. This emerged through two key dimensions: their assessment of different providers' value in patient care and their observations of how community members navigate between different types of care.

### *Value of traditional healers and faith healers as care partners*

Biomedical providers viewed faith healers as potential allies in providing holistic mental healthcare, particularly valuing their role in psychological and spiritual support. One young female provider elaborated on faith healers' contribution:

> *"These people faith healers are always hoping, praying, and counseling these people who are mentally ill. They even give these people rules*

**Table 1.** Themes inductively identified from the in-depth interviews and focus group discussions

| Theme 1: Biomedical providers' perspective on traditional healers and faith healers |
| --- |
| Sub-Theme 1: Value of traditional healers and faith healers as care partners |
| Sub-Theme 2: Observed patterns of community care-seeking |
| Theme 2: Traditional healers' and faith healers' perspectives on biomedical providers |
| Sub-Theme 1: Recognition of biomedical expertise |
| Sub-Theme 2: Experiences of exclusion |
| Sub-Theme 3: Self-perceived roles in collaborative care |
| Theme 3: Opportunities and barriers to collaboration |
| Sub-Theme 1: Economic and power dynamics |
| Sub-Theme 2: Visions for collaboration |

**Table 2.** Relationships and dynamics among providers

| Relationship | Dynamics | Observations |
| --- | --- | --- |
| Traditional healers and biomedical providers | Tension and marginalization | Power imbalances; historical colonial stigma |
| Faith healers and biomedical providers | Collaboration and complementarity | Informal patient referrals exist |
| Traditional healers and faith healers | Religious and ideological tensions | Mutual distrust rooted in differing frameworks |

*and guidance on how to live. They instill morals onto these guys. So, these people have hope and faith in God that they will be cured and healed." – IDI: Biomedical Provider, 22 year-old woman*

This positive assessment centered on faith healers' complementary role in providing psychosocial support while accepting the primacy of biomedical treatment. In stark contrast, biomedical providers expressed fundamental concerns about traditional healing practices. Their criticism often went beyond mere skepticism to active opposition:

*"That [traditional healing is fake because traditional healers say they can help, put medicine in their drugs and their herbs, and it is not measured and they end up harming them. I would not advise it and I don't like it, I wouldn't welcome a traditional healer." – IDI: Biomedical Provider, 41 year old woman*

This dismissal of traditional healing practices revealed deeper tensions about what constitutes legitimate healthcare in the current post-colonial system.

### Observed patterns of community care-seeking

Biomedical providers' accounts revealed complex patterns in how community members accessed mental healthcare. They acknowledged the significant influence of both traditional healers and faith healers in communities' initial healthcare decisions:

However, providers noted a telling distinction in how patients discussed different types of care-seeking. While patients openly acknowledged seeking help from faith healers, they often concealed visits to traditional healers:

*"When these people get these problems, some of them go to church to pray and others run to traditional healers. Many people will not open up that they have been to traditional healers. Those that go to church can tell us that they first went to church because it is not as shameful as going to traditional healers. Those who go to traditional healers first won't tell." – IDI: Biomedical Provider, 28 year old woman* A faith healer further contextualized this dynamic of patients being ashamed to admit that they went to a traditional healer:

"Traditional healers are satanic. And so being, they [patients can't easily air it out because it will seem like they trusted Satan more than God so they are afraid to say that." – IDI: Faith Healer, 45-year-old man

This pattern of disclosure suggested that colonial and post-colonial stigmatization of traditional healing practices continues to influence patient behavior and provider attitudes.

***Theme 2:*** *Traditional healers' and faith healers' perspectives on biomedical providers*

While both traditional healers and faith healers expressed respect for biomedical expertise, their relationships with the formal healthcare system revealed complex power dynamics and tensions. Their perspectives were shaped by their experiences of interaction and attempts at collaboration within the healthcare system.

### Recognition of biomedical expertise

Both traditional healers and faith healers readily acknowledged situations where biomedical care was necessary. Faith healers particularly emphasized knowing when to refer patients:

*"It is very important because once other treatments have failed, the patient with mental illness can be referred to the hospital for treatment" – IDI: Faith Healer, 59 year old man*

Faith healers generally considered biomedical care and spiritual support as complementary, and the only two forms of care that can help those with mental illness:

*"The only ways I know [to treat mental illness are two. Praying for someone according to the Lord's guidance then you give that person medicine because biomedical workers do the screening and other investigations and discover the right ingredients or chemicals that can heal the human body when there is a problem. For us in church, we believe that the evil spirits cause physical damage to the body tissues. So, it requires medicine to heal those wounds such that the person recovers." – IDI: Faith healer, 54 year old man*

Traditional healers similarly recognized that biomedical care can be helpful when traditional healing fails:

*"I think that whatever has failed me, has to be for the hospitals since people there are experienced and underwent education to handle that. So, it should be the hospitals, nothing but that." – FGD: Traditional Healer, 45 year old man*

These statements revealed a pragmatic acceptance of biomedical providers' role, particularly in cases where traditional or faith-based approaches proved insufficient. Both groups' willingness to refer patients to biomedical facilities suggested recognition of biomedicine's place in the broader mental healthcare system.

### Experiences of exclusion

While traditional healers and faith healers acknowledged biomedical providers' role, their experiences of interaction differed markedly. Traditional healers in particular described feeling marginalized and disrespected by biomedical providers. Their

accounts revealed specific instances of dismissal, particularly regarding their treatment methods:

> "When a patient comes here, we cut the skin with a razor blade and apply medicine. But if my patient reaches your facilities, they [biomedical providers despise this practice and talk a lot of words to them." – IDI: Traditional Healer, 41 year old man

The frustration with this lack of respect emerged clearly in their questions:

> "Why do [biomedical providers see that the work we do is useless? -IDI: Traditional Healer, 41 year old man

### Self-perceived roles in collaborative care

Traditional healers and faith healers viewed their roles as critical components in the care of patients with mental illness. Although they often referred patients to health centers when their treatments were ineffective, traditional healers asserted their unique expertise, particularly for conditions they viewed as having cultural or traditional causes:

> "These [biomedical providers have played a big part since mental illness is caused due to fevers and we cannot do anything about it so it is only them who can do that. Though at some times if the cause of the illness is traditional or cultural, still hospitals cannot help, it is the traditional healers that can." – IDI: Traditional Healer, 54 year old male

Meanwhile, faith healers generally viewed their roles as complementary, one even suggesting that their role was to help raise money for patients.

> "Most families are poor but if they had some money then that would be of importance in helping this patient get better, but most cases have broke families so they call faith healers to help them in raising money for treatment of their people. This is because it is hard for people to fund the treatment for people in terms of medication and transportation. – IDI: Faith Healer, 23 year old male

Faith healers also acknowledged the importance of specialized mental health services, particularly in severe mental illness, emphasizing the need for referral to national-level institutions:

> "Unless his head is paining him so much, he can be taken to Butabika [National Mental Health Hospital]. We hear that in Butabika they can check his head and give him medication and treatment." – IDI: Faith Healer, 52 years old man

The contrasting perspectives of traditional healers and faith healers revealed how colonial and religious influences continue to shape relationships with biomedical providers. While both groups acknowledged the value of biomedical care, they positioned themselves differently within the mental healthcare system. Traditional healers maintained claims of distinct expertise in treating culturally specific causes of mental illness, even as they experienced marginalization from the formal healthcare system. Faith healers, in contrast, adopted more complementary roles, focusing on spiritual support and practical assistance like fundraising. These different approaches reflected broader patterns of how each group has adapted to an increasingly biomedicine-dominated healthcare landscape – traditional healers asserting parallel but often contested authority, while faith healers found more accepted auxiliary roles within the existing system.

**Theme 3:** *Opportunities and barriers to collaboration*

The relationships between providers were shaped by deep ideological divisions, economic concerns, and power dynamics that affected potential collaboration.

### Economic and power dynamics

Traditional healers, who operated on a fee-for-service basis, expressed concerns about how collaboration might affect their livelihoods:

> "A question I have is that if we are working together, there are patients we can work on, and one party complains that they have not been paid, just as you know that always money brings problems. Yet, I cannot tell the patient because that is our secret between us. Because for you, you have to use your machines and test these patients, and I also have to use my herbs." – IDI: Traditional Healer, 31 year old man

Both traditional healers and faith healers noted power imbalances in their relationships with biomedical providers, who were seen as maintaining distance from working with the community:

> "I wouldn't be able to tell because the health workers take themselves as government workers so it is difficult for them to look for me who is on the ground and desire to collaborate with me. The only relationship we have is when I seek treatment from them." – IDI: Faith Healer, 59 year old man

### Visions for collaboration

Despite these barriers, some providers, particularly traditional healers, expressed interest in mutual learning and observation:

> "These health workers, just as that collaboration I have talked about, I would love them to come and visit me here and find out how I treat the mentally ill patients. And I should be able to go to the health facility and see how they keep the medicines and how they treat people with mental illness." – IDI: Traditional Healer, 62 year old man

Some biomedical providers perceived that there was a benefit to involving traditional healers and faith healers, as that can increase the number of patients that can be seen and referred to the health center. Biomedical providers also saw the benefit of involving faith healers in the collaborative model, as many patients trust faith healers and can improve patient outcomes.

> "It is always very easy for these people to understand and believe the words of the faith healers." – IDI: Biomedical Provider, 22 year old woman

These varying perspectives revealed how efforts to develop collaborative care models must navigate not only practical concerns but also deeply held beliefs about legitimacy, power, and the nature of mental illness treatment itself.

## Discussion

This qualitative exploratory study in rural eastern Uganda reveals the complex dynamics of traditional healers, faith healers, and biomedical providers in mental healthcare delivery. Our findings challenge the common practice of grouping traditional healers and faith healers together, demonstrating important distinctions in how these groups are perceived and how they interact with the biomedical system with consequences for developing potential collaborative models of care. Our results revealed stark differences in provider perspectives. Biomedical providers viewed faith healers as potentially helpful collaborators in providing psychosocial support but viewed traditional healers as

harmful. This negative view of traditional healers was also shared by faith healers. In contrast, both traditional healers and faith healers viewed biomedical providers as valuable potential collaborators and observed that medication could help many patients with mental illnesses when traditional treatment methods were not sufficient. These findings that traditional healers value biomedical providers as potential collaborators while biomedical providers express more ambivalence have been corroborated in other studies (Green and Colucci, 2020; Lampiao et al., 2019).

The differential treatment of faith healers and traditional healers reflects the enduring impact of colonial-era attitudes. During British colonialization, the dominant religion in Uganda became Christianity with a melding of religion and state institutions (Hansen, 1986). Colonial rulers disparaged traditional customs but allowed their use (Mamdani, 2006). However, gaining influence in the colonial state was predicated on conformity to British norms, often requiring a disavowal of traditional beliefs (Hobsbawm, 2012). A legacy of this dynamic is illustrated in our study, where service providers with access to modernity and influence (biomedical providers) have favorable views towards faith healers in an echo of the melding of religion and state during the colonial era while maintaining negative views towards traditional customs. The conflict in ideology between traditional healers and faith healers is observable, as they both expressed distrust of one another. Our study finds that collaboration between faith healers and biomedical providers is not only possible but also informally practiced with each party referring patients mutually.

However, collaboration between traditional healers and other providers are tinged with distrust, which is consistent with other literature from other African countries regarding traditional healers (Akol et al., 2018a; Burns and Tomita, 2015b; Campbell-Hall et al., 2010; Gureje et al., 2015; Musyimi et al., 2016; Sorsdahl et al., 2010). Trust is a prerequisite for successful mental healthcare collaboration (Brown and Calnan, 2016; Gilson, 2003; Illingworth, 2002). Our findings suggest that future collaboration between faith healers and biomedical providers might be easier, due to a higher degree of trust between them as many biomedical providers and faith healers share a common belief system, than between traditional healers and biomedical providers. However, it is worth noting that effective collaboration between traditional healers and biomedical providers has been achieved in other contexts, particularly in HIV care and treatment in rural Uganda (Sundararajan et al., 2021). Our findings that biomedical providers were the last line of referral for traditional healers and faith healers are also consistent with other studies (Kisa et al., 2016; Sorsdahl et al., 2010; Wagenaar et al., 2013).

Some limitations of the study warrant acknowledgment. Our study's geographic specificity to the Buyende District limits broad generalizability, though the findings resonate with broader patterns in post-colonial healthcare systems. Our research assistants, while Ugandan master's and bachelor's students, were not local to the area and represented a more educated urban demographic. Before the beginning of the study, some Ugandan research assistants expressed hostility towards traditional healing practices, which could have introduced bias. At times, the presence of American research assistants during interviews could have amplified participants' perception of an external, potentially judgmental perspective. This social positioning likely influenced participant responses, potentially reinforcing power dynamics where participants might perceive the research team as representatives of a more "modern" perspective potentially critical of traditional practices. The research team's diverse composition – including

both Ugandan and American researchers – necessitated ongoing critical reflection on how our backgrounds might influence data interpretation. We implemented strategies to mitigate potential biases, including conducting cross-cultural team discussions to challenge individual interpretations and involving at least one Ugandan and one American researcher in coding each transcript and during analysis. The presence of community health workers during initial participant introductions could have further shaped participant responses.

The results have several important implications for treatment and policy in Uganda. Our findings demonstrate the willingness of traditional healers and faith healers to collaborate and learn from biomedical providers. However, biomedical providers often viewed traditional healers and faith healers—particularly traditional healers—as unhelpful from the perspective of potential partnership. Our findings suggest that collaboration strategies with biomedical providers must be distinctly tailored to traditional healers and faith healers. Faith healers, who already occupy a more complementary role and share similar spiritual frameworks with many biomedical providers, may benefit from more integrated referral and communication systems. In contrast, collaboration with traditional healers would require more fundamental approaches that recognize their unique cultural expertise and address potential economic marginalization. Further research, particularly investigating consumer-level preferences of patients seeking healthcare, could achieve a deeper understanding of potential collaboration models and the nuanced ways patients experience and seek treatment for mental illness.

**Open peer review.** To view the open peer review materials for this article, please visit http://doi.org/10.1017/gmh.2025.18.

**Data availability statement.** The data that support the findings of this study are available from the corresponding author, YJL, upon reasonable request.

**Acknowledgements.** The team is grateful for the contribution of Buyende District community health workers, who aided in data collection.

**Author contribution.** All authors made substantial contributions to either the conception or design of the work or the acquisition, analysis, or interpretation of data. All authors drafted or revised the manuscript critically for important intellectual content and approved the final version to be published. All authors agree to be accountable for all aspects of the work and ensure that questions related to the accuracy or integrity of any part of the work are appropriately investigated and resolved.

**Financial support.** Empower Through Health provided funding for this research. Dr. Tsai reports receiving funding from US National Institutes of Health K24DA061696-01.

**Competing interests.** Dr. Tsai reports receiving a financial honorarium from Elsevier for his work as Co-Editor in Chief of the Elsevier-owned journal *SSM – Mental Health.* All other authors have no competing interests to declare.

**Ethics statement.** This research study was approved by the Institutional Review Boards of The AIDS Support Organization, Uganda (TASO-2023-222) and Yale University (2000034605). In accordance with national guidelines, we received approval to conduct the study from the Uganda National Council of Science and Technology (SS1860ES).

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
