## [Reviewer Report]

Thanks for allowing me to review this important research. The authors have done a good job and their paper could be further improved by addressing the following issues:

Abstract

State the type of qualitative design used in this study.

Mention whether the sample size was based on saturation or any other criteria.

Write the themes as presented at the start of the result section (short and straightforward).

Study design and data collection

What kind of Biomedical providers were sampled? Were they primary healthcare workers, mental health specialists, community health workers etc?

Why only one focus group for health workers and faith healers and two for traditional healers? Is there any reason or justification?

Were traditional healers a homogenous group or were there different types of healers?

Again state whether the sample size was based on saturation or another criteria.

Apart from the multiple coding, what other steps were taken to ensure the trustworthiness of the data?

Results

The sub-theme Community attitudes towards traditional healers and faith healers sound like the data comes from some community participants who hold certain attitudes. Consider revising the sub-theme.

---

## [Reviewer Report]

This is an interesting qualitative exploration of the perspectives of traditional healers, faith health and biomedical providers about mental illness treatment in Uganda. I have the following minor comments on the paper:

1. Kindly mention whether any efforts were made to triangulate the findings. If not, then it may be presented as a limitation.

2. Kindly clarify what does biomedical provider means in the present context. Does it include doctors, nurses, psychologists, etc.

3. If information is available on the duration of experience of the informant, then it may please be provided. Else do mention that the extent of experience with mental health problems may have been variable across the providers.

4. Kindly mention whether probes were used for carrying out interviews. Whether there was a list of probes developed beforehand to conduct these interviews.

5. Kindly mention the background, experience, and training of the field workers who conducted the interviews. This has implications for the assessment of reflexivity.

---

## [Reviewer Report]

General comments:

The manuscript is well written but lacks substance and critical reflection. The 18 co-authors carried out 53 interviews and 4 focus groups and report uncritically what was said, with no regard for context or positionality of interviewers. There is no discussion of the so-called “treatment gap”, the “formal mental health system” (vs an informal one). The themes overlap and are repetitive. The conclusion is that collaboration should be improved through “tailored approaches to integration” with no reference to how that might be accomplished.

Some specific comments:

(Page numbers are from the manuscript, starting with Impact Statement = page 1.)

Collaboration seems to mean persuading faith healers and traditional healers to refer people to biomedical care providers. Would the authors encourage biomedical care providers to refer patients to faith healers or traditional healers, or do they mean it should be one-way (as they later find it to be, p. 17).

By using terms such as “mental health care delivery” and “improving mental health care” (p. 1), the authors show they believe there is one global, universal form of mental health care.

The characterization of “various traditional African belief systems” is a huge generalization, not supported by at least some of the authors cited (Gbadamosi et al. 2022; Gureje et al. 2020; Gureje et al. 2015) (p. 3).

What is “mental illness” in Uganda (or the rest of Africa)? What are ¨faith healers” and “traditional healers” in the study area? (At least one biomedical participant mentions “witch doctors” (p. 3).

“Traditional healers and faith healers could potentially be brought into collaboration with biomedical providers” (p. 4), but the references cited only suggest it might be possible, not that it has ever happened.

Who were the research assistants? Were they biomedical providers? How were they trained? What was their relationship with the community? Where were the interviews carried out?

One of the sub-themes is “Community attitudes towards traditional healers and faith healers” (p. 9). But the authors didn’t interview community members; instead, they cite biomedical providers for this theme.

“Biomedical providers generally acknowledged the beneficial role of faith healers in holistic patient care” (p. 9). But the authors also say (p. 16) that “Many biomedical providers … were often religious,” and that “Most biomedical providers believed that medication was the primary treatment, while prayer and counseling was an optional additional treatment.” Not surprising, if they are religious/Christian. (I think this is tied to the “the lasting impact of colonial-era attitudes” they refer to.)

“Some providers noted that many patients felt ashamed and embarrassed about seeing traditional healers.” (p. 11). That is quite understandable if the interviewers were biomedical providers or associated with them, or were local and known to be Christian, and community health workers were present.

“When a patient comes here, we cut the skin with a razor blade and apply medicine” (p. 13). So not just mental illness, but applying medicine for fevers, etc.

The authors mention “society stigma face by traditional healers.” According to whom?

---

## [Reviewer Report]

General comment

The authors have explored important public health issue and the manuscript is presented well. Though the analysis still needs further analysis using thick description (breadth, depth, context and nuance). Please find bellow some specific feedback

1. The study design is not mentioned or not clear

2. Why both FGDs and IDIs were used needs explanation

3. What strategies the authors employed to ensure rigor of the study?

4. What qualitative data analysis software was used for the data management?

5. No any data visualization efforts?

6. The analysis lacks adequate breadth, depth, context and nuance. for example, the authors can present some of the findings based on the perceived severity or type of mental illness. The authors could probe the findings based on severity and they just presented as if all mental illness are the same

7. The terms “many” and “some” patients or respondents can be characterized.

8. were there no participant who is both faith and traditional healer?

9. There is a statement on the result section “No faith healers endorsed referral to traditional healers. Likewise, no traditional healers reported referring patients to faith healers”. What does it mean? I am sure all the traditional healers have some faith and do not believe in prayer. Is it not contrasting finding? or where is the discrepancy?

---

## [Editor Report]

Three reviewers recommend revisions and on reviewer recommended rejection. All reviewers have very substantive comments that require major revision. In particular more detail is required on the methodology as well as reporting on the findings. 

Please attend to all four reviewers' comments and re-submit.

---

## [Editor Report]

Publication is recommended for acceptance. The authors‘ have responses to the reviewers’ comments are acceptable. Just for noting by the authors for future publications is to strive to be more balanced with respect to co-authors from high-income and co-authors coming from the LMIC country where the research is conducted as well as to strive towards having a local LMIC lead author.